# Perceptions on Extending the Use of Technology after the COVID-19 Pandemic Resolves: A Qualitative Study with Older Adults

**DOI:** 10.3390/ijerph192114152

**Published:** 2022-10-29

**Authors:** Ceci Diehl, Rita Tavares, Taiane Abreu, Ana Margarida Pisco Almeida, Telmo Eduardo Silva, Gonçalo Santinha, Nelson Pacheco Rocha, Katja Seidel, Mac MacLachlan, Anabela G. Silva, Oscar Ribeiro

**Affiliations:** 1Digital Media and Interaction Research Centre (DigiMedia), Department of Communication and Art, University of Aveiro, 3810-193 Aveiro, Portugal; 2Center for Health Technology and Services Research (CINTESIS@RISE), Department of Education and Psychology, University of Aveiro, 3810-193 Aveiro, Portugal; 3Governance, Competitiveness and Public Policies (GOVCOPP), Department of Social, Political and Territorial Sciences, University of Aveiro, 3810-193 Aveiro, Portugal; 4Institute of Electronics and Informatics Engineering of Aveiro (IEETA), Department of Medical Sciences, University of Aveiro, 3810-193 Aveiro, Portugal; 5ALL Institute, Department of Anthropology, Maynooth University, W23 F2H6 Maynooth, Ireland; 6Assisting Living and Learning Institute (ALL Institute), Department of Psychology, Maynooth University, W23 F2H6 Maynooth, Ireland; 7Center for Health Technology and Services Research (CINTESIS@RISE), School of Health Sciences, University of Aveiro, 3810-193 Aveiro, Portugal

**Keywords:** technology, COVID-19, older adults, digital solution, eHealth

## Abstract

The COVID-19 pandemic of the last two years has affected the lives of many individuals, especially the most vulnerable and at-risk population groups, e.g., older adults. While social distancing and isolation are shown to be effective at decreasing the transmission of the virus, these actions have also increased loneliness and social isolation. To combat social distancing from family and friends, older adults have turned to technology for help. In the health sector, these individuals also had a variety of options that strengthened eHealth care services. This study analyzed the technologies used during the COVID-19 pandemic by a group of older people, as well as explored their expectations of use after the pandemic period. Qualitative and ethnographic interviews were conducted with 10 Portuguese older adults, and data were collected over a period of seven months between 2020 and 2021. The research demonstrated that the use of current and new technologies in the post-pandemic future is likely to be related to overcoming: (i) insecurity regarding privacy issues; (ii) difficulties in using technologies due to the level of use of digital technology; and (iii) the human distancing and impersonal consequences of using these technologies.

## 1. Introduction

In the last two years, the lives of many individuals have been affected by the severe acute respiratory syndrome coronavirus 2 (SARS-CoV-2) pandemic [1,2,3], also known worldwide as COVID-19 [4,5]. COVID-19 has affected groups that are more vulnerable and at higher risk of exposure, infection, and death, such as older adults. To prevent the spread of the virus, several countries adopted social distancing, isolation, and quarantine as important public health measures [6]. Moreover, great efforts were made to establish and reinforce e-Health and telehealth geriatric care services [7,8] with increased availability of e-consultations and e-prescriptions.

Although the actions taken were found to be effective at decreasing the transmission of SARS-CoV-2, they could also increase unwanted loneliness and social isolation [9], with the potential to affect both physical and mental health [10,11]. Unwanted loneliness and social isolation can occur at any stage of an individual’s life, but they increased particularly during the pandemic among older adults, often aggravating pre-existing situations of social and emotional vulnerability. To combat social distancing of family and friends, many older adults turned to technology for help, especially communicative technologies, such as WhatsApp^®^, Skype^®^, and Zoom^®^ [11,12,13]. Through individual or group conversations, older adults were able to maintain contact with family members and long-time friends, with several socioemotional benefits [14,15,16]. In Portugal, the country that this study focuses on, the use of these social networks stood out as mental and physical health tools. From social networks, individuals were able to stay informed about the pandemic situation [17], keep in touch with people close to them to combat loneliness and social distancing [18], and perform physical exercises inside the comfort of their homes to maintain good physical health [11].

Within the health sector, the use of technological options increased greatly during the pandemic, from consultations through voice and video calls to patient tracking applications, remote triage emergency services, chatbots, diagnostic tools with artificial intelligence (AI), voice interface systems, and mobile sensors, such as smart watches, oxygen monitors, and thermometers, among others [19,20]. A review intended to synthesize the capabilities of m-Health in providing health services to the older population during the COVID-19 pandemic showed an overall positive effect, with most of them used for therapy, information provision, self-help, monitoring, and mental health consultation [21]. According to this review, m-health applications kept both older adults and healthcare providers safe, accelerated health provision, reduced costs of service provision, and decreased the risk of morbidity and mortality [21]. Moreover, remote consultations proved to be significant ways to support non-severe COVID-19 patients and ensure access and continuity of care for non-COVID-19 patients, leading to a rapid expansion in the use of digital tools in many countries in Europe [22]. In Portugal, on the other hand, remote consultations, and different e-Health solutions available to the population proved to be below expectations. A review points out the requirements, potentials, constraints, and aspects of possible improvements of the various e-Health solutions adopted in the country [23]. According to this review, Portugal still needs to create solid foundations in the digital architecture of healthcare solutions to reach the user needs in relation to privacy and security, interoperability, and inclusion [23].

Despite the growing volume of literature reporting on how the COVID-19 pandemic favored the adoption of digital technologies for socialization purposes [24] and within geriatric healthcare [19], few studies have yet reported on the expected continuity of use of digital technologies by older adults when the pandemic resolves. Knowing older adults’ attitudes, opinions, and motivations to learn and/or continue using technologies is of crucial importance for planning the adoption of continuous enrolment in training and lifelong learning offers related to technological usage, and, therefore, contributes to active and healthy aging and ensure that the potential of technologies is achieved.

This study analyzed the use of technologies during the COVID-19 pandemic by a group of older adults, as well as explored their expectations of use after the pandemic period. The focus was on 10 Portuguese cases with different life situations (e.g., in terms of levels of education and digital technology usage), who were interviewed during a seven-month period, between 2020 and 2021, within the scope of an ongoing European project entitled SHAPES (Smart and Healthy Aging through People Engaging in Supportive Systems), aimed at developing a Digital Platform and Ecosystem to make digital solutions and services for an aging population available [25].

## 2. Materials and Methods

In a previous work within SHAPES, the “lifeworld” of a sample of older Europeans was investigated through qualitative and ethnographic interviews in ten pilot and reference sites [26]. This particular study focuses on the interviews carried out by Portuguese researchers in Portugal. Each participant was interviewed at least twice from August 2020 to February 2021. The plan for this research went through the Ethics Committee on Social Sciences at Maynooth University (SRESC-2021-2428941) and all interview participants gave their informed consent to participate in the research.

The selection of participants followed a list of sociocultural and demographic categories that sought to reach as much as possible a varied spectrum of life situations, including distinct living contexts/situations (e.g., alone or with a partner), diversity in digital skills, and different health status (e.g., people with serious and chronic health conditions or mobility restrictions, and autonomous, independent healthy older adults).

Except for one case where the interview was conducted face-to-face, the interviews were conducted remotely through Zoom^®^ meetings due to COVID-19 security restrictions, with one or two researchers. The interviews lasted between 30 min and 2 h each and followed a pre-defined semi-structured interview guide developed at Maynooth University addressing a set of themes including the experience and impact of the coronavirus pandemic, family and social life, use of technology, and health, care, and wellbeing [26]. Participants were interviewed at least twice each, to answer all the questions in the interview guide. The variability of time in each interview moment reflects each participant’s willingness to explore the interview topics.

The sample was composed of older adults living in the community and the participants were characterized in terms of age, gender, living situation, education, digital technology usage, and internet access at home. The participants’ digital technology usages were defined according to the answers obtained from the questions about the theme of the use of technology: low digital technology usage (participants who only used the computer with the help of family members); medium digital technology usage (participants who knew how to use instant messaging applications (such as WhatsApp^®^), as well as consult e-mails and news sites); high digital technology usage (participants who knew how to use the technologies mentioned on the level before and also use video call applications, such as Zoom^®^, control their finances through mobile applications, and could overall easily adapt to new technologies).

The interviews were recorded and transcribed verbatim in the original language (Portuguese) and then translated into English. The researchers also took notes during interviews. The transcript interviews and notes were analyzed through content analysis using the qualitative data analysis software WebQDA^®^ by two of the authors (C.D. and O.R.) who grouped the transcribed text into themes and categories with similar meanings [27]. The categories for the initial analysis emerged from the reading and review of the interviews and notes, with the objective of identifying themes relevant to the study. From the identification of these initial categories, it was defined that the focus of this content analysis would be on the use of technology. The resulting findings, patterns, and relevant topics that surfaced from the two themes and four categories were clustered and cross-analyzed.

## 3. Results

A total of 11 Portuguese older adults participated in the study, but only 10 older adults were considered for the final sample (as one of the participants did not agree to sign the consent form): six females and four males with a mean age of 73.81 (SD 1.96) years old. Nine participants lived in predominantly urban areas, and one in a rural area. Eight lived with a partner and two lived alone. Participants had different levels of education, being divided into university education (*n* = 5), high school, i.e., twelve years of schooling (*n* = 2), primary/compulsory education, i.e., approximately eight years of schooling (*n* = 2), and basic education, i.e., one to four years of schooling (*n* = 1). Regarding digital technology usage level, most (*n* = 6) had high degrees, and two had low degrees. Still, in relation to the digital environment, eight participants had access to high-quality internet at home; the other two had no internet access in their homes. A summary of the participants’ characteristics is presented in Table 1.

Nine interviews were conducted remotely via video conference, and one took place at the participant’s home. To adapt the interview process to the needs of the participants, nine participants were interviewed on two different occasions, and one took place in over three different occasions, totaling 21 interview sessions and more than 22 h of recording time. Details about each of the interviews are presented in Appendix A.

Regarding the content analysis of the interviews and notes, all transcribed statements were aggregated into three main themes: (i) Use of technologies during COVID-19, (ii) difficulties, challenges, and fears of technology use, and (iii) future technology usage.

The first theme was divided into three categories: (i) for managing social distancing; (ii) for health purposes; (iii) for daily life. The first category (for managing social distancing) presents comments on the strategies adopted by older adults to overcome isolation and social distancing from family and friends. In this category, there are mainly exposed examples of the use of video calling applications to chat with family members and to hold work meetings, instant messaging groups with friends, and e-mails for distant friends who do not use certain technologies. The second category (for health purposes) mainly involves the use of technology to track news about the pandemic (e.g., the daily count of new cases and the number of deaths from COVID-19), and the use of technologies for online medical consultations. eHealth solutions have become common during the pandemic due to the risk of contagion when traveling to medical offices and hospitals. The third category (for daily life) is related to the use of technologies for financial matters (e.g., checking bank accounts, paying bills, and deposits, among others). Mandatory isolation has prevented older adults from going to banks and ATMs to resolve financial matters, making home banking the main solution during the pandemic.

The second theme (difficulties, challenges, and fears of technology use) addresses the concerns of older adults when using current and new technologies. As barriers to the use of technologies, older adults emphasize privacy and security issues, and the fear of using technologies they do not know—either due to the low level of digital technology use or lack of interest. This topic also presents comments related to the distance in the relationship and contact between doctors and patients based on the use of eHealth technologies.

The third theme (future technology usage) is related to the perception of older adults regarding the use of new technologies in the post-pandemic period. Some participants pointed out the desire to continue using technologies after the pandemic, as well as interest in using new technologies, even suggesting options that they would consider useful in their daily lives. Examples of citations that support these themes and categories are presented in Table 2.

## 4. Discussion

This study analyzed the use of technologies during the COVID-19 pandemic by a group of older adults and explored their expectations of use after the pandemic resolves. It allowed reinforcing already available insights on how older adults related to technologies during the COVID-19 health crisis and how they used them (or reinforced their use) as instruments to overcome the pandemic restrictions, but it also presented information on how the possible continuity of technology use is perceived. Complimentarily, the study shed some light on the challenges, uncertainties, and worries that older adults face when using technologies.

In general, it is widely acknowledged that the pandemic generated several barriers to older adults, mainly since they were the age group with the highest risk of contamination by the virus and mortality [16]. One of the most striking barriers during this period was social distancing and isolation. This issue has been studied by several researchers, who show that home confinement has a negative effect on social participation and the life satisfaction of individuals [28,29], and generate greater symptoms of psychological distress, which usually persist for a long period after the end of the isolation period [30]. These aspects negatively impact the quality and life experiences of older adults, who depend on access to family and friends, emotional support, mutual help in daily life, higher levels of education, and interpersonal intimacy, among others [31]. As an option for improving the quality and experience of life in this period, technology stands out, which provides personal meaning, a sense of agency and self-management, self-growth, as well as other developments centered on old age [32].

In this context, as a strategy to combat social distance from family and friends, manage daily life matters from home, and access health care services, older adults were (to some extent) forced to embrace digital life, adapting themselves to a new life context and showing signs of resilience [33,34]. The technologies were mostly used to maintain social networks (reducing feelings of loneliness), practice exercise routines at home [11], stay informed about world events (especially those related to COVID-19), and manage daily life tasks (mainly financial matters). For participants, technology use during this period was a big change in terms of previous routines that were typically conducted through face-to-face interactions. In this specific group of Portuguese participants, who were all aged between 70 and 77 years old, most of them presented a medium/high degree of digital technology usage and felt, to some extent, comfortable in handling technologies. Nevertheless, they expressed reliance on the help of family members to sustain a proficient use of basic tablet functionalities (e.g., video conferencing). This was, in fact, one transversal topic in all themes/categories that echoes previous research on the challenges older adults often face when handling new technologies, including dependence on family members for technical support [35]. This implies that those with less locally available family support, who may in fact benefit more from communication technology, may find it harder to access them. 

It is important to mention those participants in which digital technology usage was low. Studies suggest that gender, age, and education level are the most critical factors influencing technology use [36,37], and this was apparent in two female participants who had relatively low education levels. Both participants only used technology to maintain contact with family and friends through instant messaging or video call applications and needed the help of partners or family members to be able to use and configure these applications. One of the participants presenting low digital technology usage lived in a rural area. Unlike the others, she had no interest in technology and refused to acquire it or learn how to use it. The reasons given for adopting this attitude included a lack of trust in technologies, especially smartphones; and a belief that these technologies were unnecessary for her lifestyle.

Regardless of the “forced entrance” into a greater digital life or at least to an upscaled use of technology, the recurrent use during the pandemic period contributed to developing skills and enhancing the level of digital technology usage. However, it also contributed to a clearer perspective (and a more critical one) about the challenges and difficulties technology use brings. Along with the need for someone to help in using (and/or teach) specific devices and applications, there was also the suggestion that a comfortable relationship with technology is an important need to assure older adults, with several participants expressing fears and concerns on privacy and security matters, especially in social networks and online banking. These concerns have already been reported in previous studies, which point out that older adults reject digital technologies because they are afraid of making mistakes [38], or think they are not capable of using them [39], and because they do not see the benefits in the use of technologies [36]. Other studies highlight that older adults do not use technologies for reasons of privacy, cost of acquisitions, self-efficacy, and fear of dependence [40,41,42].

In relation to health-related technologies, despite considering that the care received during the pandemic period was of good or excellent quality, respondents also presented several doubts and reservations about the use of health technologies to fulfill their needs. This was mainly due to the lack of human contact between health professionals and patients, a reality also pointed out in the literature by mental health professionals, who found it difficult to work with patients’ emotional behaviors due to the lack of human contact [43]. In fact, although all were in favor of using e-Health in such adverse circumstances (lockdown), it was not clear the extent to which participants favored the continuity of its use after the pandemic. On this matter, a recent European research report [44] on older adults’ lives during the COVID-19 pandemic and how they were impacted by governments and societal responses revealed that although low-tech e-healthcare facilitated access to healthcare, about 56% of people aged 50+ who needed a consultation had a face-to-face consultation because they preferred it to the available e-healthcare options—an e-healthcare consultation did not fully meet the needs of 49% of people aged 50+ who used one. Other authors had already reported how the pandemic crisis put the spotlight on the limitations of technologies in terms of meeting older adults’ needs, stating that electronic contact may not have the same potential as face-to-face contact to reduce the negative impacts of the pandemic situation, at least in mental health [45].

Overall, the set of limitations, challenges, and fears presented by the ten cases included in this study may be directly related to the possibility of using technologies in the future. While they have fears about the safety of technology and difficulties using it, they also hope that in the post-pandemic future new technology will be developed that may help them with everyday issues and problems, which have become even more evident during the confinement period. One identified problem/concern is having free Wi-Fi everywhere and for everyone, which was mentioned by some participants who, despite reporting having good coverage of the internet at home, underlined the limitations of not having an affordable generalized connection to the internet outside of their homes.

The lack of access to the internet, constraining the use of web-connected technological devices, as well as the barriers to using technology in an independent manner (which may fall into the so-called digital divide) are currently crucial matters of concern for governments worldwide that need to recognize that citizenship, in addition to social and economic participation, must mean digital participation [8]. This must be extended to those who are today labeled as the “oldest old” (i.e., aged 80 and over), and who, despite being an understudied group in terms of technology use, are presented as everyday users [46]. In fact, though an age-related digital divide may still exist, older adults are increasingly crossing it and becoming users of a range of technologies [47]. However, technology training and digital infrastructures are necessary if the full potential of technology is to be achieved in a time that is shaped by an unprecedented “longevity revolution”, which recognizes lifelong learning as a crucial active aging pillar [48]. Overcoming difficulties in using/learning how to use technology by the older generations may strongly depend on the available help of third parties (i.e., family, friends, or professionals) or whether there will be opportunities for lifelong learning. Ensuring supported access and training should, therefore, be guaranteed. These issues all resonate with the importance of promoting a social justice approach to ensuring that access to digital infrastructure is fair and considering the different barriers faced by different groups in society [49].

With this study, it was noticed that in Portugal, despite being a country with a large part of its population composed of older adults (23.4% are 65+ years old, according to the provisional 2021 census [50], the use of technology by older adults still faces barriers and limitations. Most Portuguese older adults do not have high levels of digital technology usage, internet access, and new technologies [51,52]. It can be seen from the comments of participants that technological use is still heavily based on instant messaging and video calling applications. When questioned about the use of new technologies, most do not know about potentially relevant technologies, other than those they already use, and do not necessarily see the need to explore the opportunities associated with new ones. This demonstrates that despite finding in our results opportunities for the continued use of technology in the post-pandemic period, there are still challenges to be considered, such as internet access, a low level of digital technology usage, acceptance of new technology, the learning process inherent to (start) using technology in advanced ages, and concerns regarding privacy security. These factors warrant a deeper investigation of the results presented in this qualitative study; their salience (and approach) to addressing them may be different for older persons than for younger persons. 

## 5. Limitations

The interviews were carried out primarily in an online format due to the COVID-19 pandemic. As pointed out by several participants, face-to-face meetings reinforce human contact between the interviewer and participant, and this lack of contact may have influenced the way participants answered the questions, often providing superficial and straightforward answers to issues that could be addressed in more detail if conducted in person. Moreover, conducting interviews in an online format in Portugal limited the number of participants in this study, because according to research data [51,52], in 2021, although 71% of individuals aged between 55 and 64 years used the internet, this number decreased to 47.7% in individuals aged between 65 to 74 years, being expectedly lower after age 75. Another limitation concerns the possibility of diversifying sample profiles. Although the participant selection variables allowed a wide range of combinations of possible interviewee profiles, it was not possible to have participants of each type of profile. Another limitation was the inclusion of a small number of participants from a rural area, who mostly do not have computers and/or internet access in their homes, and who have low levels of digital technology usage. These factors also limited the participants’ sample diversification. For future studies, it will be important to expand the sample of participants.

## 6. Conclusions

During the COVID-19 pandemic, individuals have seen modern technology rapidly evolve, marked primarily by the implementation of new digital technologies with massive speeds and impacts. Despite the rapid growth of these new technologies and the increase in the amount of information available in the digital space, many older adults still do not have sufficient knowledge of how to use them. As can be seen from the participants included in this study, the use of current and new technologies in the post-pandemic future is dependent on overcoming issues related to (i) insecurity regarding privacy issues; (ii) difficulties in using technology due to the level of digital technology usage, and (iii) the human distance and impersonal treatment that these technologies provide. Although there were participants who intend to continue using current technologies in the future, and others who anticipate using new technologies of interest to them, for others, the obstacles identified by using such technologies may be greater than the desire to use them.

## Figures and Tables

**Table 1 ijerph-19-14152-t001:** Summary information of cases.

Case	Age	Gender	Environment	Living Situation	Education	Digital Technology Usage	Access to Internet at Home
Case 1	74	Male	Urban	With partner	Primary/compulsory	High	Yes
Case 2	73	Male	Urban	With partner	University	High	Yes
Case 3	73	Male	Urban	With partner	University	High	Yes
Case 4	73	Female	Urban	With partner	University	High	Yes
Case 5	73	Female	Urban	With partner	High School	Medium	Yes
Case 6	72	Female	Urban	With partner	High School	Low	Yes
Case 7	72	Female	Urban	Alone	University	High	Yes
Case 8	70	Female	Rural	Alone	Primary/compulsory	Other	No
Case 9	75	Female	Urban	With partner	Basic	Low	No
Case 10	77	Male	Urban	With partner	University	High	Yes

**Table 2 ijerph-19-14152-t002:** Themes, categories, and illustrative citations resulting from the content analysis of the interviews and notes.

Themes	Categories	Citations Examples
Use of technologies during COVID-19	for managing social distancing	“I really think this is an extraordinary thing [referring to video meetings], because being here at home and watching my grandson play, dance… dance, or something like that! I couldn’t do any of this a few years ago!” (Case 2). “[being part] of a golf group, we started sending messages… we are about 40… messages from one, messages from another… extremely interesting things, you know? Jokes, politics, everything… an immense universe! That lead me to keep myself connected to my friends, it was a way to keep the connection (…) WhatsApp was a tool that allowed me to keep connections, to continue connections” (Case 3). “people who do not forget us, and of whom we also do not forget [on keeping connected virtually]…, especially those friends who are very lonely, who became widows… (…) whenever I can I send messages, also emails… one of them [friend] is practically blind and he cannot see well on his cell phone, [so] connection has to be by email so that he can augment the words” (Case 4). “Due to the pandemic, Alberto usually speaks with his work colleagues using a video call app. He also uses it to meet monthly with the members of the civic think-tank and record the meetings” (Fieldnote—Case 10).
	for health purposes	“I follow some information about the COVID-19 pandemic through the internet” (Case 1). “[on having a phone call with his doctor] I did one recently, due to this pandemic the doctor contacted me […] He was very careful, although he was not the doctor that usually accompanies me, he was very careful, he made a true consultation, he elucidated me [on the pandemic], he informed me, he informed me in a way that I appreciated”. (Case 1). “I count the days as I’m following through the internet the numbers [of infected cases and deaths], the numbers are dropping each day (…) I spoke with the doctor on the phone, I told him what I needed, and he immediately sent me the prescription, and I didn’t even mention it! It was all very good” (Case 2).
	for daily life	“I check my bills; I check the electricity bills (…) now I do all through the tablet” (Case 1). “(…) I use the internet, but not only for information—I have the bank there, I have the internet there, I have the finances there, I have everything, I consult everything around” (Case 2). “I use the computer, I analyze the banking balances, check if there has been any movement or not, I see the news, what is happening in the world (...)” (Case 3)
Difficulties, challenges, and fears of technology use		“Whenever I have problems [using technologies], I ask my daughter or my son-in-law, they’re the ones helping me out regularly. I do need help.” (Case 1). “[about new technologies and applications] I don’t have a computer myself, I have a tablet, which is this one where we are talking through, but yes, for me that’s enough (laughs) and that’s how I started to adapt (…) I think that this is enough—WhatsApp, the internet… I have Google, my Gmail, my email, so I think that this is enough“ (Case 1). “I just don’t know if it’s safe [to be in social networks], the insecurity for me is very big (...) my children have their own channels to talk to their mother and something like that, channels of family groups… but I don’t know if it’s safe, honestly… But you don’t put anything wrong there either (…) but I find it very insecure (…)” (Case 2). “[about phone calls with the doctor] It’s different when we’re talking directly to the doctor, we look in the eyes like this [illustrating facially] and sometimes we ask more questions, it’s different from doing it over the phone, it’s totally different, for me it’s different… I’m not sure of its use in the future…“ (Case 2). “[about video calls with her doctor] it does not have the same human warmth; it is completely different…” (Case 7). “Isabel stressed how she does not trust technology. She understands that technology is important and that it helps many people, but she believes that she would feel somehow trapped if using a cell phone. She claims that she is happier and freer without technology, the reason why she only has a television, a radio, and a landline phone” (Fieldnote—Case 8). “I don’t know if I ‘d use [referring specifically to a smartwatch], I’m sincere, I don’t like clocks, time is given to me for free, I occupy it the way I can, the best way I can, I don’t run with anything, I don’t worry about anything, I get far, I get closer, I don’t know if I would use it, no, no. It would bring me worries in my head, stress, it’s better to live in my calm, without all that.” (Case 8).
Future technology usage		“[about the use of new smartwatches] I bought this, but it cost me 55 euros, which is almost nothing, there are much more sophisticated ones than this, but that’s what I said, “it suits my needs…”. So, I bought this, and I walk with it, I see sleep, I see how the heart rate is evolving, which at some point can help a little with these health problems that I intend to monitor. This is something I’ll keep using“ (Case 3). “I would like a central heating, I don’t have it, I have heaters in the rooms and everything, but it’s different, now carry it down or have a boiler, for my house to be all warm… [about the use of a platform that control biometric data] I think you would be comfortable, because we never know what will happen next, but I trust in such a situation of giving my data and not being violated, right?“ (Case 5). “I think that there should be free Wi-Fi everywhere so we don’t have to spend so much money and so we all can have access to everything.” (Case 7). “[about new technologies and applications with specifically functions] I think that the oximetry function, if there was a way, maybe there is… If there was a way to have a permanent assessment of the levels of oxide, which I am guaranteeing for my blood during the day to day, it would be useful. It gave me rest and I think it would even allow me to regulate my activity more effectively. I don’t know if it even exists, but if it doesn’t, it would be nice to have it.” (Case 10).

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
