# Peer review of "Perceptions on Extending the Use of Technology after the COVID-19 Pandemic Resolves: A Qualitative Study with Older Adults"

_ijerph, 2022, doi:10.3390/ijerph192114152_

Round 1
Reviewer 1 Report
The authors report the results of a qualitative research on the intriguing topic of the use of technology after the COVID-19 pandemic in ten older adults. The article is well written and clear in every section. The Introduction clarifies the background of the research; the methods section makes it easier for the reader to understand the interview procedure and data collection. The Results follow the usual reporting rules for this type of research and are appropriately deepened in the discussion. The conclusions are appropriate to the reported results. I have just a few detailed comments:
1 I suggest to include in the title the description of the type of study: Perceptions on extending the use of technology after the COVID-19 pandemic resolves: a qualitative study with older adults
2 There are no data on how many refused to participate or abandoned the study
3 I could not find the reference 21.
I suggest that you indicate if you have considered the COREQ checklist ( Tong, A., Sainsbury, P., & Craig, J. (2007). Consolidated criteria for reporting qualitative research (COREQ): A 32-item checklist for interviews and focus groups. International Journal for Quality in Health Care, 19(6). https://doi.org/10.1093/intqhc/mzm042 )
Reviewer 2 Report
line 28: define COVID-19
line 34: change analyses to analyze
line 47: risk for what? Exposure, infection, death, impacting the health of others? Be specific and relate it to your topic.
Line 52: define SARS CoV-2 and connect it to COVID-19.
Line 60: the study took place in Portugal but the sample size is small: dies it represent a particular geographic area or demographic group?
It is unclear how the 10 participants were chosen. Although their commonalities and differences are well documented, why stop at 10? Show your calculations of how a sample size of 10 impacts the reliability of the data (and conclusions). Do they represent a small area of Portugal or are they from 10 different regions? How representative of Portugal are these 10 subjects?
Are their attitudes a result of the pandemic or did they always have these attitudes?
Methods: what were the questions that were asked? How did the 2 interviews differ? How were they the same? What types of questions were asked? Most subjects were interviewed for about 2 hours yet there are few themes. Did you pick and choose topics or was this the only topic asked about?
The Discussion shows that you are well aware of the literature but you have chosen a very small number of subjects to study. The subject group is diverse, which makes it difficult to make general statements about how the impact of different characteristics. The last paragraph of your Discussion might be better placed in the Introduction in a effort to focus the Introduction more. The Introduction is too long and not focused.
Reviewer 3 Report
I read the paper entitled "Perceptions on extending the use of technology after the COVID-19 pandemic resolves: a study with older adults". It is a very interesting and original study.
First of all, congratulations to the authors for the excellent manuscript they submitted. I believe that this paper should be published after minor revision.
My comments follow:
-In the title, I believe that the authors should state that this is a qualitative study and that this study concerns Portuguese older adults
- I may be exaggerating, but the older adult's reference to branded search engines and branded apps reminded me of advertising.
-I would ask the authors to separate the Appendix into a different file.
Reviewer 4 Report
Dear authors, thank you very much for the opportunity to read your paper. I think it will be interesting for many readers. However, given the method chosen, I find the paper lacking a deeper interpretation of the categories. Your results chapter is descriptive, not interpretive, which is a great pity. I would recommend the results chapter to be much more detailed and in-depth.
Reviewer 5 Report
Dear authors:
First of all, I would like to congratulate you on the work carried out and the chosen subject matter. Next, I would like to suggest some areas for improvement:
- The abstract should be structured in which you provide background, methods, results and conclusion.
- Line 50, references 2,3 should be joined [2,3]. Same line 53, 58, 70 and the whole text.
- The section on material and methods should be expanded further, as it is not clear how patients were included and how they were interviewed, whether by Zoom or Whatsapp. In addition, they should include that the study complies with the Declaration of Helsinki, the sample size calculation and a flow chart with the inclusion and loss of patients.
- In table 1 they should remove the pseudonymisation of patients.
- Table 2 does not include any results, but only refers to the questions asked. It would be good to include it as an annex, but not in the results section.
- The conclusions section is not decisive and should provide a conclusion with more force and not so superficial.
In general, the article should improve its wording, especially the section on methods and results. The sample of the study is very poor and in my opinion cannot show significant evidence, as well as the bias of interviewing patients makes the study inconclusive and without evidence.
Regards.
Round 2
Reviewer 2 Report
Thank you for considering making changes,
Reviewer 4 Report
The deeper interpretation of the data is still missing. It is not evident what are the key findings.
Reviewer 5 Report
Dear authors:
Thank you for the amendments made to the document. The main concern was the ethical aspect and you have solved it, however, in my view the article lacks an adequate sample and therefore the conclusions cannot be supported in this statistical study with such a small and biased sample. I suggest that you present the article with a larger sample to obtain real data.
Regards
